# Influence of Flow Rate Distribution on Combustion Instability of Hypergolic Propellant

**Yushan Gao [1], Bingbing Zhang [2], Jinbo Cheng [2], Jingxuan Li [2,3] and Qingfei Fu [2,3,*]**

1   Science and Technology on Liquid Rocket Engine Laboratory, Xi'an Aerospace Propulsion Institute, Xi'an 710100, China
2   School of Astronautics, Beihang University, Beijing 100191, China
3   Ningbo Institute of Technology, Beihang University, Ningbo 315100, China
*   Correspondence: fuqingfei@buaa.edu.cn

**Abstract:** Combustion instability is the biggest threat to the reliability of liquid rocket engines, whose prediction and suppression are of great significance for engineering applications. To predict the stability of a combustion chamber with a hypergolic propellant, this work used the method of decoupling unsteady combustion and acoustic system. The turbulence is described by the Reynolds-averaged Navier–Stokes technique, and the interaction of turbulence and chemistry interaction is described by the eddy-dissipation model. By extracting the flame transfer function of the combustion field, the eigenvalues of each acoustic mode were obtained by solving the Helmholtz equation, thereby predicting the combustion stability for the combustion chamber. By predictions of the combustion chamber instability with different flow rate distributions, it was found that the increasing of inlet flow rate amplitude will improve the stability or instability of combustion. The combustion stability of the chamber was optimized when the flow rate distribution for the oxidant was set more uniform in the radial direction. The heterogeneity of the flow rate distribution in the circumferential direction is not recommended, considering that a homogeneous flow rate distribution in the circumferential direction is beneficial to the combustion stability of the chamber.

**Keywords:** liquid rocket engine; flow rate distribution; thermoacoustic decoupling; combustion instability

## 1. Introduction

Liquid rocket engines (LRE) have been widely used in spacecraft launch and recovery due to their following advantages: high specific impulse, high thrust, the ability to start repeatedly, variable operating time, adjustable thrust, and repeated usage [1]. Combustion instability is widely found in rocket engines, aero engines, large air heaters and so on, which will lead to uncontrollable, huge pressure pulsation in the combustors. This pulsation can cause backfire, flameout, and overheating of the combustor wall, which may lead to a series of problems, such as shortening the engine life and even destroying the engine [2]. Predicting and suppressing the combustion instability of the combustor in LRE is of great significance to the development of aerospace industry.

In order to reduce the occurrence of combustion instability in experiments, theoretical and numerical simulation methods are generally used to predict the combustion instability. Prediction methods of combustion instability are divided into two categories [3].

The first kind is directly solving the coupled combustion instability system by computational fluid mechanics and compressible solver, which can obtain the frequency of combustion instability. Leonardi et al. [4] implemented a specific module based on the double time lag model and the coupling of the combustion chamber, and feed line oscillations were investigated by using a complete set of nonlinear equations. This method generally uses large-eddy simulation (LES) to obtain the information of the coupled combustion instability system. Chen et al. [5] studied self-exciting combustion instability in twin scroll GTMC (gas turbine model combustors) and its interaction with air-fuel mixing by the LES

technique. The detached eddy simulation (DES) method was developed in recent years and has been used to the prediction of combustion instability because its advantages in the computational efficiency of Reynolds-averaged Navier–Stokes (RANS) and high accuracy of LES. Yuan et al. [6] captured self-exciting high-frequency combustion instability in a rocket engine combustion chamber by the two-dimensional DES technique and verified the existence of self-exciting high-frequency combustion instability.

The second kind decouples the combustion system and acoustic system and calculates the flame response and acoustic response with the incoming flow rate disturbance, respectively. Generally, the acoustic response is calculated by a low-order acoustic model, and the flame response is represented as the transfer relationship between the combustion heat release rate and incoming flow rate disturbance, such as the flame transfer function. Urbano et al. [7] simulated a full-scale combustor by the LES technique and studied the combustion instability of BKD with the acoustic modal identification method. The study showed that the instability of first-order tangential and first-order longitudinal plays an important role in the combustion instability of the chamber. Li et al. [8] used the LES technique to simulate the combustor and substituted the transfer function into OSCILOS, which is a combustion instability prediction and simulation program, and finally obtained accurate prediction values of thermoacoustic oscillation frequency and amplitude.

Influence factors of combustion instability were studied by experimental and numerical simulation methods to suppress the occurrence of combustion instability. There are many factors that affect combustion instability, such as Reynolds number (Re), combustion chamber geometry [9], mixing ratio, type of fuel, degree of premix and turbulence intensity [10]. By affecting the heat release rate and its coupling with acoustic waves, these factors cause different combustion instability phenomena in combustors.

Experimental research studies on combustion instability can be traced back to the 1950s. A lot of research has been carried out on the combustion instability of LRE. The F-1 rocket engine used in the Apollo moon landing program has undergone abundant experiments so as to solve the combustion instability problem. By costing billions of dollars and testing thousands of combustion chamber geometries, the program accumulated plenty of experience in combustion instability research [11]. Bazarov et al. [12] found the injector structure influence on combustion instability. The dynamic characteristics of injectors were analyzed theoretically and experimentally. At present, experimental methods are mainly based on two types of advanced optical testing methods: the particle image velocimetry (PIV) and planar laser-induced Fluorescence (PLIF), which can measure the process data and then research on combustion instability. Soller et al. in Germany [13,14] studied the oxygen/kerosene coaxial swirl combustor. They observed the longitudinal high-frequency combustion instability phenomenon during the experiment and concluded that the injector structure is the key factor on combustion dynamics. Wang et al. [15] conducted experimental studies on a single-injector engine. The length of injector recess influenced combustion instability, but the influence of the combustion chamber length on the longitudinal high-frequency combustion instability was more obvious. Xue et al. [16] carried out experiments on a single-injector rectangular combustion chamber, which showed that there is a relatively optimal recess ratio to make the combustion more stable under supercritical conditions. Bai et al. [17] conducted an experimental study on a combustor with liquid-centered coaxial swirling injectors, and their research showed that when the self-exciting oscillation frequency of the injectors coupled with the natural acoustic frequency of first order longitudinal mode (1L) of the combustor, the combustor pressure would oscillate at the same frequency. This phenomenon suggested that self-exciting oscillation may be a key factor in the combustion instability of LRE. Armbruster et al. [18] conducted experiments on combustion instability on the BKD combustion chamber. The research showed that the self-exciting oscillation of injectors makes the combustion chamber transform from the first-order tangential mode, with a larger amplitude, to the first-order longitudinal mode, which is lower frequency and couples with the self-exciting oscillation frequency. Stefan et al. [19] conducted an in-depth study on the coupling of self-exciting oscillation

and combustion instability, which showed that the flow oscillation of injectors has a greater impact on the heat release rate change than sound pressure. Laera et al. [20] proposed an experimental device that can detect combustion stability and combined the self-exciting oscillation experiment with numerical simulation to study the effect of a single-injector combustion chamber length on the longitudinal combustion instability [21,22]. Hardi et al. [23] took the BKH combustion chamber as an example to study the interaction between the sound field and combustion field, experimentally. The study showed that the oscillation of the sound velocity seriously affects jet breakup when it gets close to the jet velocity.

In numerical simulation, Fureby [24] simulated combustion instability by the LES technique, and the simulation results showed that vortex shedding is an important factor causing unsteady heat release. Nie [25], Cheng [26] and Feng [27] conducted a lot of simulation research on the injector structure parameters and physical parameters of hypergolic propellant engines, hydrogen oxygen engines and hydrocarbon engines, and concluded that structures such as baffles can suppress high-frequency combustion instability. Qin et al. [28] found that, compared with the short and thick combustion chamber, the elongated chamber is beneficial to suppress tangential mode combustion instability which is the most harmful factor to the engine. Fang et al. [29] conducted a series of studies on combustion chamber with liquid oxygen/methane injectors and studied the influence of injector structure parameters on the atomization angle and combustion performance experimentally and numerically. Kraus et al. [30,31] predicted combustion instability considering wall heat transfer by the LES technique. The results showed that the heat transfer of wall has a certain impact on combustion field simulation. Wu et al. [32] simulated the engine with $UDMH/N_2O_4$ and compared with the experimental results of the hydrogen oxygen engine. The results showed that the two engines have similar variation trends of temperature and Mach number, but the tail flame core temperature of $UDMH/N_2O_4$ engine is relatively low.

In this paper, the decoupling thermoacoustic system method is adopted. The flame transfer function is obtained by computational fluid dynamics, while the acoustic mode is solved by Helmholtz equation, and the combustion instability of a combustion chamber is predicted. The influence of the flow rate amplitude and flow rate distribution on combustion instability is explored by studying different conditions.

## 2. Combustion Instability Prediction Method

This chapter introduces the combustion instability prediction method used in this paper. The combustion field and the acoustic system were decoupled, and the combustion response caused by acoustic perturbations is characterized as the flame transfer function. The transfer function can be obtained experimentally or numerically. In this paper, it is difficult to obtain the flame transfer function by experimental means, so the numerical method is adopted. The prediction of combustion instabilities is accomplished by substituting the resulting flame transfer function into the Helmholtz equation. The two key steps are obtaining the flame transfer function by simulating the combustion field and solving acoustic modes by the Helmholtz equation, which are described in Sections 2.1 and 2.2, respectively.

### 2.1. Combustion Field Numerical Model

This section describes the methods used to obtain the flame transfer function. The combustion field distribution is obtained by computational fluid dynamics methods, mainly using the eddy-dissipation model, $k - \omega$ SST turbulence model and Redlich–Kwong equation of state. The flame transfer function is extracted from the combustion field and described by the $n - \tau$ model.

2.1.1. Governing Equations

(1)    Continuity Equations

When solving mixed flows containing different substances, the percentage of each substance in the mixture can be obtained by solving the convection–diffusion equation of each substance. This conservation equation follows the following form:

$$\frac{\partial}{\partial t}(\rho Y_i) + \nabla \cdot (\rho \vec{v} Y_i) = -\nabla \cdot (\vec{J_i}) + R_i \tag{1}$$

where $Y$ is the mass fraction of the component, and $R$ is the rate of the component produced by the chemical reaction. In a multicomponent flow with N components, N-1 equations in Equation (1) need to be solved. Usually, to minimize numerical errors, the transport equation corresponding to the other components is solved instead of solving for the component with the largest share in the mixture. $\vec{J_i}$ is the diffusion flux of the $i$-th component, resulting from the concentration and temperature gradient of the component. For laminar flow, the diffusive flux is defined as follows:

$$\vec{J_i} = -\rho D_{i,m} \nabla Y_i - D_{T,i} \frac{\nabla T}{T} \tag{2}$$

where $D_{i,m}$ is the mass diffusion coefficient of the $i$-th component in the mixture, and $D_{T,i}$ is the corresponding temperature diffusion coefficient. For turbulent flow, the diffusive flux is defined as

$$\vec{J_i} = -(\rho D_{i,m} \nabla Y_i + \frac{\mu_t}{Sc_t}) - D_{T,i} \frac{\nabla T}{T} \tag{3}$$

where $Sc$ is the Schmidt number and $\mu$ is turbulent viscosity.

(2)    Momentum Equation

The momentum conservation equation in the inertial reference frame is as follows:

$$\frac{\partial}{\partial t}(\rho \vec{v}) + \nabla \cdot (\rho \vec{v} \vec{v}) = -\nabla \cdot p + \nabla \cdot (\bar{\bar{\tau}}) + \rho \vec{g} + \vec{F} \tag{4}$$

where $p$ is the static pressure, $\tau$ is the pressure tensor, and $\rho g$ and $F$ are the gravitational field and the external force field, respectively. The pressure tensor $\tau$ is given by

$$\bar{\bar{\tau}} = \mu[(\nabla \vec{v} + \nabla \vec{v}^T) - \frac{2}{3} \nabla \cdot \vec{v}] \tag{5}$$

(3)    Energy Equation

$$\frac{\partial}{\partial t}(\rho E) + \nabla \cdot (\vec{v}(\rho E + p)) = -\nabla \cdot (\sum_j h_j J_i) \tag{6}$$

(4)    Equation of State

Under the pressure and temperature conditions of the combustion chamber, the propellant is in a transcritical or supercritical state. At supercritical pressure, since the fluid properties change continuously from the jet to the surrounding environment, using traditional methods to deal with the material properties will introduce particularly large errors. The errors become particularly huge especially as the fluid approaches the critical point. Therefore, to characterize the physical properties of supercritical fluids more accurately, it is necessary to develop a physical property evaluation mechanism that is applicable to the entire thermodynamic state region, especially the combustion chamber studied in this paper, which is in the supercritical state.

The Redlich–Kwong (R-K) equation is an effective method for this system. It fits better in the supercritical state and has smaller errors than the Soave–Redlich–Kwong (SRK) and Peng–Robinson (PR) equations and is simpler than the other correction equations.

Therefore, the R-K equation is used to describe the physical parameters in this simulation. Its expression is as follows [33]:

$$p = \frac{R_g T}{v - b} - \frac{a}{T^{0.5} v(v + b)} \tag{7}$$

where $p$ is the pressure, $T$ is the temperature, and $R_g$ is the universal gas constant. $a$, $b$ are the parameters of the equation, which can be specifically expressed as [33]:

$$a = 0.4278 R_g{}^2 T_c^{2.5} / p_c \tag{8}$$

$$b = 0.0867 R_g T_c / p_c \tag{9}$$

where $p_c$ is the critical pressure of the substance and $T_c$ is the corresponding critical temperature.

### 2.1.2. Turbulence Model

The $k - \omega$ SST turbulence model is used in this paper, which is developed from the $k - \omega$ model but has higher accuracy and confidence in a wide range of flow domains, and it is best suited for the simulation of the combustor. The transport equation for $k - \omega$ model is as follows:

$$\frac{\partial}{\partial t}(\rho k) + \frac{\partial}{\partial x_i}(\rho k u_i) = \frac{\partial}{\partial x_j}\left[\Gamma_k \frac{\partial k}{\partial x_j}\right] + G_k - Y_K \tag{10}$$

$$\frac{\partial}{\partial t}(\rho \omega) + \frac{\partial}{\partial x_i}(\rho \omega u_i) = \frac{\partial}{\partial x_j}\left[\Gamma_\omega \frac{\partial \omega}{\partial x_j}\right] + G_\omega - Y_\omega \tag{11}$$

where

$$\begin{aligned} \Gamma_K &= \mu + \frac{\mu_t}{\sigma_K}; \\ \Gamma_\omega &= \mu + \frac{\mu_t}{\sigma_\omega}; \\ \mu_t &= \alpha^* \frac{\rho k}{\omega}; \\ \mu_t &= \rho C_\mu \frac{k^2}{\varepsilon}; \end{aligned} \tag{12}$$

In Equations (10)–(12), $G_k$ represents the turbulent kinetic energy generated by the average velocity gradient; $G_\omega$ represents the amount of $\omega$ produced; and $\Gamma_K$ and $\Gamma_\omega$ are the effective diffusivities of $k$ and $\omega$, respectively. $Y_K$ and $Y_\omega$ are the dissipation rates of $k$ and $\omega$ due to turbulence, respectively.

Compared with the $k - \omega$ model, the improvement of $k - \omega$ SST is that the effect of the turbulent shear stress is taken into account when defining the turbulent viscosity term. This makes it more accurate and reliable in simulating the separation of airflow from smooth surfaces as well as cases with wall-restricted flow.

The turbulent viscosity expression for the above-mentioned model is as follows:

$$\begin{aligned} \mu_t &= \frac{\rho k}{\omega} \frac{1}{\max\left[\frac{1}{\alpha^*}, \frac{SF_2}{a_1 \omega}\right]} \\ F_2 &= \left(\Phi_2{}^2\right) \\ \Phi_2 &= \max\left[2\frac{\sqrt{k}}{0.09 \omega y}, \frac{500\mu}{\rho y^2 \omega}\right] \end{aligned} \tag{13}$$

### 2.1.3. The Eddy-Dissipation Model (EDM)

In certain operating conditions, the fuel burns rapidly, and the overall reaction rate is dominated by turbulent mixing. For example, in a high-temperature non-premixed flame, the turbulence causes the fuel and oxidant to mix slowly in the reaction zone and burn quickly. In this case, the following assumption is made: combustion is controlled by fuel mixing, ignoring the complex chemical reaction rates, and assuming that the rate of combustion is much greater than the mixing rate (complete combustion once mixed).

Based on the assumption of "combustion occurs once mixing" proposed by Magnussen and Hjertager [34], ANSYS Fluent has a built-in reaction model based on turbulence control

called the eddy-dissipation model. In this model, the generation rate of a substance is defined as follows:

$$R_{i,r} = \frac{v'_{i,r} M_{w,i} A \rho}{\tau} \left( \frac{Y_{\Re}}{v'_{\Re,r} M_{w,\Re}} \right)$$

$$R_{i,r} = \frac{v'_{i,r} M_{w,i} AB \rho}{\tau} \left( \frac{\sum_P Y_P}{\sum_j^N v''_{j,r} M_{w,j}} \right) \tag{14}$$

The meanings of each term in the above formula are as follows: $Y_p$ is the mass fraction of the product; $Y_{\Re}$ is the mass fraction of the reactant; and $A$ and $B$ are the empirical coefficients, which are set as 4.0 and 0.5, respectively.

In Equation (14), the chemical reaction rate is controlled by the time scale of large eddy mixing, defined as $\tau$. As long as $\tau > 0$ is established, the reaction will proceed, with no ignition source required to start the combustion reaction. Xu et al. [35] used the eddy-dissipation model to simulate the combustion of hypergolic propellants, MMH/NTO, and it is feasible to use this model to simulate the combustion of hypergolic propellants, which makes us choose the eddy-dissipation model.

The combustion chamber calculated in this paper is a rich fuel environment: the total mixing ratio of the combustion chamber (0.2051), and the maximum mixing ratio of the two-component injector (0.6488), are lower than the stoichiometric ratio of the complete reaction of UDMH/$N_2O_4$ (3.0616, can be calculated by Equation (15)). Therefore, the total package reaction model with a mixing ratio of 1.53078 is chosen in this paper, as expressed in Equation (16).

$$C_2H_8N_2 + 2N_2O_4 = 2CO_2 + 3N_2 + 4H_2O \tag{15}$$

$$C_2H_8N_2 + N_2O_4 = CO_2 + 2N_2 + H_2O + CO + 3H_2 \tag{16}$$

2.1.4. Flame Transfer Function

The flame transfer function used in this paper is defined as

$$F_j(t) = \frac{q'_j}{u_r'} \tag{17}$$

where $q'_j$ is the combustion heat release rate pulsation, and $u'_r$ is the velocity pulsation of the reference point. Since the prediction is performed in the frequency domain, Equation (17) is Fourier transformed and becomes the following expressions:

$$\hat{F}_j(f) = n_j e^{i\varphi_j}$$

$$n_j = \frac{\hat{q}_j}{\hat{u}_r} \tag{18}$$

where $\hat{q}_j$ is the perturbation amplitude of $q'_j$ at frequency $f$, and $\varphi_j$ is the phase difference between $q'_j$ and $u'_r$. Since $u'_r$ is also a disturbance parameter that needs to be calculated, this section only considers the treatment of the heat release rate. The function expression was first proposed by Crocco [36]. The flame transfer function relates the relationship between the unsteady heat release rate and the incoming flow rate disturbance, thus avoiding the complex description of the relationship between flow, acoustics and combustion. Hence the transfer function between inlet mass flow rate perturbation and combustion heat release rate is introduced:

$$H_j(t) = \frac{q'_j}{\dot{m}'/\overline{\dot{m}}} = \frac{q'_j}{\alpha \sin(2\pi f t)} \tag{19}$$

2.2. Thermo-Acoustic Decoupling Method

The theoretical rough estimation method and Helmholtz equation are used to solve the decoupled sound field. Xiang [37] used the method introduced in this section to predict combustion instability in long flame combustor. By comparing with the results of the direct

coupling solution under the same conditions, it shows the feasibility of the decoupling method used in this paper. Section 2.2.1 introduces the application conditions of decoupling and the theoretical rough estimation method. Section 2.2.2 introduces the principle of the Helmholtz equation to solve the decoupled sound field.

### 2.2.1. Preliminary Estimation of Acoustic Modes

There are two methods for the rough estimation of the acoustic modes: (1) theoretical prediction; and (2) solving the Helmholtz equation of the steady field without the according source term. For the former, the value of the thermodynamic parameters is an average value, regardless of the complex combustion chamber structure, while the latter method is more accurate. In this paper, the steady field is used first to predict the frequency, and then the initial value is substituted into the Helmholtz equation to solve.

The geometry of this combustion chamber is relatively simple, with no variation in the cross-sectional area of the combustion chamber section, which is cylindrical in shape, but with a constricted segment nozzle downstream. Due to the large flow velocity in the nozzle, the Mach number is generally greater than 0.1, so it cannot be considered a non-mean flow. Since the sum of the injector area is quite different from the cross-sectional area of the combustion chamber, and the temperature difference between the injectors and the combustion chamber is relatively large, the acoustic system upstream of the injectors could be decoupled from that in the combustion chamber [2]. The general decoupling criteria is as follows: $\frac{S_2}{S_1}\sqrt{\frac{T_2}{T_1}} > 10$, where subscript 2 represents the combustion chamber, and subscript 1 represents the injectors. The main body of this model is the combustion chamber with nozzle shown in Figure 1. The nozzle is treated as an acoustic impedance boundary condition using the corresponding boundary condition. In the simulation, only the front part is considered, shown in Figure 2, whose acoustic modes need to be calculated. When preliminarily estimating the frequency of the acoustic modes in the combustion chamber, the injector inlet can be regarded as an acoustic closed condition, and the outlet can also be regarded as a closed condition.

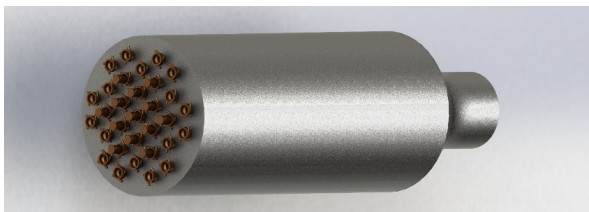

**Figure 1.** Structure of the combustion chamber.

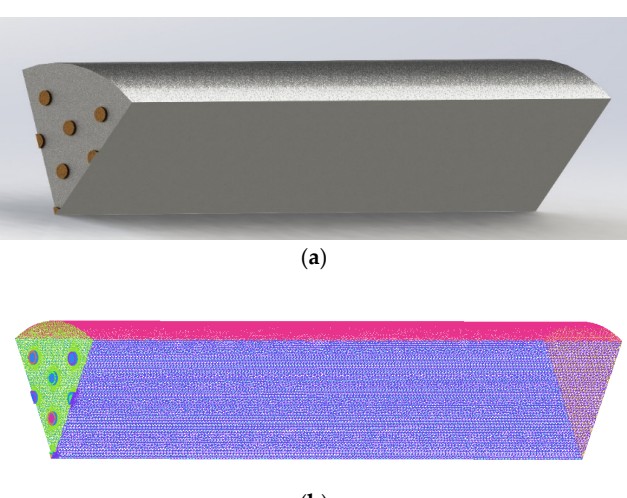

(**a**)

(**b**)

**Figure 2.** The 1/6 combustor structure. (**a**) Computational domain. (**b**) Computational grid.

The frequencies of each acoustic modes (symbolized as $f_{nmj}$) can be roughly estimated according to the classical formula:

$$f_{nmj} = \frac{c_c}{2\pi} \sqrt{k_r{}^2 + k_{r,n}{}^2} = \frac{c_c}{2\pi} \sqrt{4\left(\frac{\alpha_{mj}}{D_c}\right)^2 + k^2{}_{x,n}} \tag{20}$$

where $n$, $m$, $j - 1$, $C_c$, $k_r$, $k_{x,n}$, $\alpha_{mj}$ and $D_c$ are the longitudinal mode number, the tangential mode number, the radial mode number, the sound velocity of the combustion chamber, the radial wave number, the longitudinal wave number, the wave number perpendicular to the longitudinal direction, and the diameter of the combustion chamber, respectively, where

$$k_{x,n} = \frac{n\pi}{l_c} \tag{21}$$

For a cylindrical shape, $\alpha_{mj}$ is generally the root of the Bessel function $\frac{dJ_m(\alpha_{mj})}{dr} = 0$, which can be calculated by looking up the Table 1 as follows.

**Table 1.** The value of $\alpha_{mj}$ corresponding to each mode.

| $\alpha_{mj}/2\pi$ | $m = 0$ | $m = 1$ | $m = 2$ | $m = 3$ |
|---|---|---|---|---|
| $j = 1$ | 0 | 0.2930 | 0.4861 | 0.6686 |
| $j = 2$ | 0.6098 | 0.8485 | 1.0673 | 1.2757 |
| $j = 3$ | 1.1166 | 1.3586 | 1.5867 | 1.8058 |
| $j = 4$ | 1.6192 | 1.8631 | 2.0961 | 2.3214 |

### 2.2.2. Helmholtz Equation

The Mach number basically does not exceed 0.1 in the combustion chamber in this case, so the average flow is ignored in the acoustic calculation of this combustion chamber, and only the pressure disturbance is considered. The Helmholtz equation can be established for the entire combustor, with the flame transfer function regraded as part of the source term. By coupling the Helmholtz equation and the flame transfer function, using the aforementioned acoustic boundary conditions of the nozzle, and reasonably setting the acoustic boundary conditions at the inlet, the acoustic modes of the entire system can be obtained.

For a region with a heat source, the fluid density $\rho$ varies with pressure $p$ and entropy $s$. By the derivation rules for compound function,

$$\frac{D\rho}{Dt} = \frac{1}{c^2}\frac{Dp}{Dt} + \frac{\partial\rho}{\partial s}\Big|_p \frac{Ds}{Dt} \tag{22}$$

Combining the linearized mass and momentum conservation equations, the equation of the combustion chamber can be written as

$$\frac{1}{\bar{c}^2}\frac{\partial^2 p'}{\partial t^2} - \bar{\rho}\nabla \cdot \left(\frac{1}{\bar{\rho}}\nabla p'\right) = \frac{(\gamma - 1)}{\bar{c}^2}\frac{\partial q'}{\partial t} \tag{23}$$

This inhomogeneous equation is the wave equation used to describe the pressure perturbation $q'(\mathbf{x}, t)$ caused by the unstable heat input, which is the core formulation when using COMSOL. In fact, the wave equation used in COMSOL is the Helmholtz equation, which is expressed as

$$\nabla \cdot \left(-\frac{1}{\rho_0}(\nabla p - \mathbf{q})\right) - \frac{\omega^2}{\rho_0 c^2}p = Q^{CM} \tag{24}$$

This formula is slightly different from Formula (23). Based on the above two equations, the unipolar domain source can be solved simultaneously:

$$Q^{CM} = \frac{i\omega(\gamma - 1)}{\overline{\rho}\overline{c}^2} q' \tag{25}$$

With the equation $q' = Q'/V$, where $V$ is the volume of the heat source, Equation (26) can be obtained by substituting it into Equation (25):

$$Q^{CM} = \frac{i\omega(\gamma - 1)}{\overline{\rho}\overline{c}^2} \frac{Fu'_r}{V} \tag{26}$$

Based on the momentum conservation equation $\partial \mathbf{u}'/\partial t + \nabla p'/\overline{\rho} = 0$ and $f(t) = Re[f_{max} exp(i\omega t)]$, it can be derived that

$$u'_r = -\frac{\nabla p'_r}{i\omega\overline{\rho}} \tag{27}$$

Substitute into the previous formula to obtain

$$Q^{CM} = -\frac{\gamma - 1}{\overline{\rho}^2\overline{c}^2} \frac{F}{V} \nabla p'_r \tag{28}$$

According to the FLUENT simulation calculation results, several assumptions are set: the flame length is defined by the highest temperature point at the central axis of the injector; the gradient of the heat release along the flame axis is ignored; the flame is regarded as a compact one. Based on these assumptions, $n$ and $\varphi$ were calculated corresponding to different injectors, substituted into COMSOL for simulation calculation.

## 3. Physical Model and Boundary Conditions

### 3.1. Physical Model

The combustion chamber structure studied in this paper is shown in Figure 1, which mainly includes a cylindrical combustion chamber and a nozzle. The injection panel consists of 13 coaxial swirling injectors and 18 single-component fuel swirling injectors. The locations of the coaxial swirling injectors are as follows: 7 large flow rate injectors in the central zone and 6 small flow rate injectors in the outer ring.

Since the injectors distribution is centrosymmetric, the combustion field can be simplified by periodic boundaries. In this work, for different flow rate distribution schemes, part of the combustion chamber is used for combustion simulation, according to the symmetry characteristic. As shown in Figure 2, the structure and the corresponding unstructured grid of 1/6 combustion chamber are used for the simulation. The number of grids was 1.7 million. The 1/3 combustion chamber is also used in some flow rate distribution schemes.

### 3.2. Boundary Conditions

There are several considerations before making assumptions: (a) it is difficult to realize the coupling method, which considers the continuous cold state phase in the injectors and the high temperature combustion field at the same time; (b) for the application of the discrete phase method and the volume of fluid method, the model establishment of droplet evaporation under high pressure environment has not formed an industry-recognized standard, and computation amount increases greatly as well; (c) for the pressure and temperature in the combustion chamber, the propellants are in a supercritical state. Therefore, the assumptions for the inlet state are as follows: UDMH and $N_2O_4$ are treated as continuous phase; these physical parameters (such as density) are fitted by the R-K equation, while the transport parameters are solved by the method corresponding to the state. Continuous phase assumption is effective for solution while maintaining correctness.

Based on this assumption, the general schemes for simulation are as follows: the combustion chamber inlet is set from 0.5 mm upstream of each injector recess; the axial and tangential velocity of the two propellants at the injector recess are obtained according to the theoretical calculation of Zhang [38]; and the boundary condition for inlet is set as the velocity inlet, while the outlet is set as pressure outlet. The specific velocity is given in Section 3.3.

When simulating the steady combustion field, the inlet velocity is a constant value obtained by the above method. To obtain the flame transfer function, a sinusoidal disturbance of a certain frequency is applied to the axial velocity of the inlet so as to achieve an effect similar to the sinusoidal disturbance of the inlet mass flow rate. The form of the inlet mass flow rate is shown in Equation (29).

$$\dot{m} = \overline{\dot{m}} + \dot{m}' = \overline{\dot{m}} \cdot (1 + \alpha \sin(2\pi f t)) \tag{29}$$

The frequency in Equation (29) is given according to the rough estimation of the acoustic modes in Section 2.2.1.

### 3.3. Scheme Settings

To explore the influence of the mass flow rate amplitude and flow rate distribution on the combustion instability of the hypergolic propellant combustion chamber, all the schemes are obtained by making minor adjustments to the basic scheme of the combustion chamber in this section. For the basic scheme, the entire combustion chamber is composed of 31 injectors, characterized into three types: (a) 18 single-component fuel injectors in the outermost ring; (b) 6 coaxial swirling injectors with a small oxidant flow rate whose mixing ratio is 0.302, located in the middle ring; and (c) 7 coaxial swirling injectors with a large oxidant flow rate whose mixing ratio is 0.649, located in the center. The schematic diagram is shown in Figure 3.

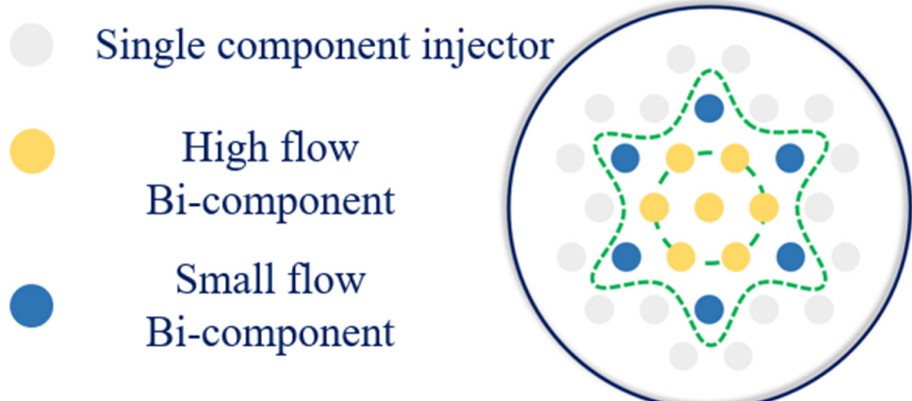

**Figure 3.** Injector distribution in the injection panel.

### 3.3.1. Inlet Flow Rate Oscillation Schemes

In order to obtain the flame transfer function, this section sets a case to verify which propellant the oscillation should be applied to and the amplitude in Equation (29). For the question of whether the oscillation should be applied to the oxidant circuit or the fuel circuit, two types of oscillation schemes are used in the basic case. The oscillation is set in the oxidant circuit in Scheme 1, and the oscillation of the fuel circuit is set in Scheme 2. As for the amplitude, 10%, 20%, and 30% oscillations are respectively applied to the oxidant circuit in the basic scheme, corresponding to Schemes 1, 3, and 4. The schemes introduced in this section are shown in Table 2.

**Table 2.** Settings for Schemes 1 to 4.

| | Flow Rate Distribution | Oscillation Amplitude | Oscillation Circuit | Research Question |
|---|---|---|---|---|
| Scheme 1 | Basic flow rate distribution scheme | 10% | oxidant | Location of oscillation |
| Scheme 2 | | 10% | fuel | |
| Scheme 3 | | 20% | oxidant | Effect of amplitude |
| Scheme 4 | | 30% | fuel | |

### 3.3.2. Flow Rate Distribution Schemes

To explore the influence of flow rate distribution on combustion instability, various flow rate distribution schemes are set in this section. All of them are obtained by modifying the basic flow rate distribution scheme, which are described in Section 3.3. The flow rate distribution schemes in this section keep the total flow rate and mixing ratio in the entire combustion chamber unchanged, with only the local flow rate distribution varied.

In order to explore the influence of the radial propellant flow rate distribution, the flow rate scheme is set according to the following rules: the total flow rate and mixing ratio of the injectors in each circle are kept the same as the basic flow rate distribution, but the flow rate of each injector in each circle is changed; the flow rate of two adjacent injectors increases and decreases, respectively, and the amplitude of all injectors' flow rate changes is equal. A sketch map of the flow rate change is shown in Figure 4. The basic flow rate distribution of the basic injector structure, the 5% circumferentially variable flow rate distribution, and the 10% circumferentially variable flow rate distribution scheme are respectively explored, which are set as Schemes 5 and 6, respectively. The scheme settings and flow rate of typical injectors are shown in Table 3.

In order to explore the influence of the oxidant radial flow rate gradient, the flow rate scheme is set according to the following rules: the fuel flow rate of all injectors is fixed, and in each circle, the flow rate of each injector is kept the same; and the oxidant flow rate of the central 7 bi-component injectors and the oxidant flow rate of the six bi-component injectors in the outer ring are changed, as shown in Figure 5. According to Table 4, three different flow rate distributions are used, and the table also shows the flow rate of the typical injectors.

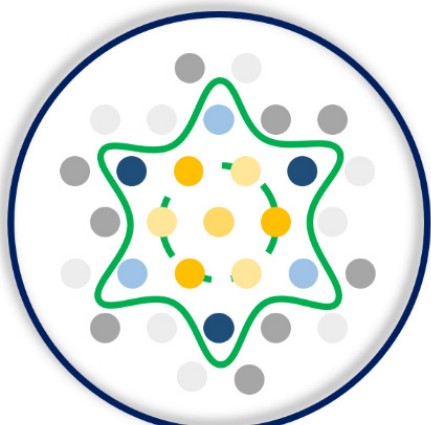

**Figure 4.** Sketch map of flow rate change in Scheme 5. The dark color indicates large flow rate, while the light color indicates small flow rate.

**Table 3.** Settings for Schemes 5 and 6.

| | Scheme 1 | Scheme 5 | | Scheme 6 | |
|---|---|---|---|---|---|
| | | **Large Flow** | **Small Flow** | **Large Flow** | **Small Flow** |
| Inner ring oxidant injector flow rate (kg/s) | 0.228 | 0.24 | 0.216 | 0.25 | 0.206 |
| Outer ring oxidant injector flow rate (kg/s) | 0.106 | 0.112 | 0.1 | 0.116 | 0.096 |
| Fuel injector flow rate (kg/s) | 0.352 | 0.368 | 0.3336 | 0.386 | 0.316 |

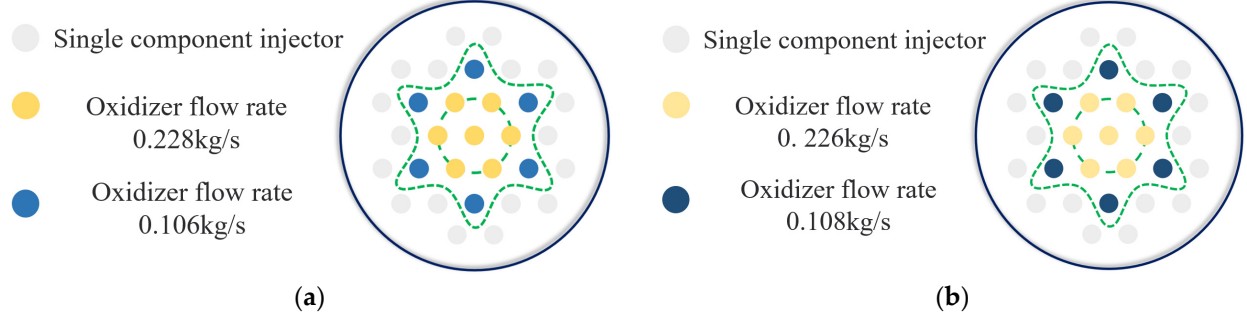

(**a**)            (**b**)

**Figure 5.** Example of flow change. (**a**) Scheme 1. (**b**) Scheme 7.

**Table 4.** Setting for Schemes 7 and 8.

| | Scheme 1 | Scheme 7 | Scheme 8 |
|---|---|---|---|
| Inner ring oxidant injector flow rate (kg/s) | 0.228 | 0.226 | 0.23 |
| Outer ring oxidant injector flow rate (kg/s) | 0.106 | 0.108 | 0.104 |
| Inner ring oxidant total flow rate (kg/s) | 1.594 | 1.584 | 1.604 |
| Outer ring oxidant total flow rate (kg/s) | 0.638 | 0.646 | 0.628 |
| Oxidant flow rate ratio of inner and outer rings | 2.50 | 2.45 | 2.55 |

## 4. Results and Analysis

### 4.1. Influence of Inlet Flow Rate Oscillation on Combustion Instability

According to the rough estimation method of the combustion chamber acoustic mode introduced in Section 2.2.1, the steady state frequency of each scheme is roughly estimated by using the steady field. The steady fields of the four examples in this section are the same. The average total temperature of the combustion chamber is 1500 K, the average sound velocity is 600 m/s, the diameter of the combustion chamber is 150 mm, and the length is about 400 mm. Putting the above parameters into Equation (20), the theoretically predicted eigenfrequency of each acoustic mode can be obtained as shown in Table 5.

According to the hot test, the dangerous mode is first-order longitudinal, zero-order tangential and zero-order radial mode, so the theoretical predicted frequency of this mode (869 Hz) is set as the initial iterative value for solving the Helmholtz equation without the source term. The temperature, sound velocity and density distribution of the steady field are interpolated and imported into the equations, and the actual characteristic frequency of the combustion chamber under the steady state field is 1280.6 Hz. To calculate the flame transfer function, the oscillation frequencies for Schemes 1 to 4 are chosen close to their characteristic frequency, that is, 1000 Hz, 1300 Hz, and 1600 Hz. Add oscillation at the same time, extract the heat release rate distribution of each injector within 2 ms, and calculate the flame transfer function. Based on these, the instability prediction of combustion can be obtained by solving the Helmholtz equation.

**Table 5.** Rough estimation of each modal frequency of the combustion chamber.

|  | Zero-Order Tangential, Zero-Order Radial | First-Order Tangential, Zero-Order Radial | Second-Order Tangential, Zero-Order Radial | Zero-Order Tangential, First-Order Radial |
|---|---|---|---|---|
| Zero-order longitudinal | 0 | 3191 | 5294 | 6641 |
| First-order longitudinal | 869 | 3307 | 5364 | 6697 |
| Second-order longitudinal | 1738 | 3633 | 5572 | 6864 |
| Third-order longitudinal | 2607 | 4120 | 5901 | 7134 |

### 4.1.1. Influence of Oscillation Circuit on Combustion Instability

There are two feedback mechanisms in the processes of inducing combustion instability. The first is to apply the flow rate disturbance to the injector inlet, which causes oscillation inside the injector, inducing combustion instability in the combustion chamber. Disturbance of the oxidant circuit or fuel circuit causes combustion instability, which is obviously a positive feedback process. The second one is disturbance feedback from the combustion chamber to the injectors, and the feedback process is represented by the velocity pulsation value. Which circuit oscillation has a greater impact depends on the second feedback mechanism, that is, the feedback brought by from combustion oscillation to the interior of the two types of injectors. The strength of this feedback can be characterized by the velocity pulsation value.

It can be seen from Figure 6 that, for Schemes 1 and 2, no matter where the flow rate disturbance is applied, the feedback from the combustion process to the inside of injectors is always that the velocity pulsation value of the oxidant circuit is larger. When the disturbance is added to the oxidant circuit for simulation, the reference velocity taken by the heat source in the acoustic simulation is correspondingly larger, which gives a larger heat source, and it is more likely to cause the oscillation of the heat release rate in the combustion chamber, further stimulating combustion instability. While the disturbance is applied to the fuel circuit, on the contrary, it is given a smaller heat source, and when the stability of the acoustic mode is further calculated and predicted, even if the result is stable, combustion instability may occur. Therefore, adding the flow rate disturbance to the oxidant circuit can ensure the correctness of the results to the greatest extent.

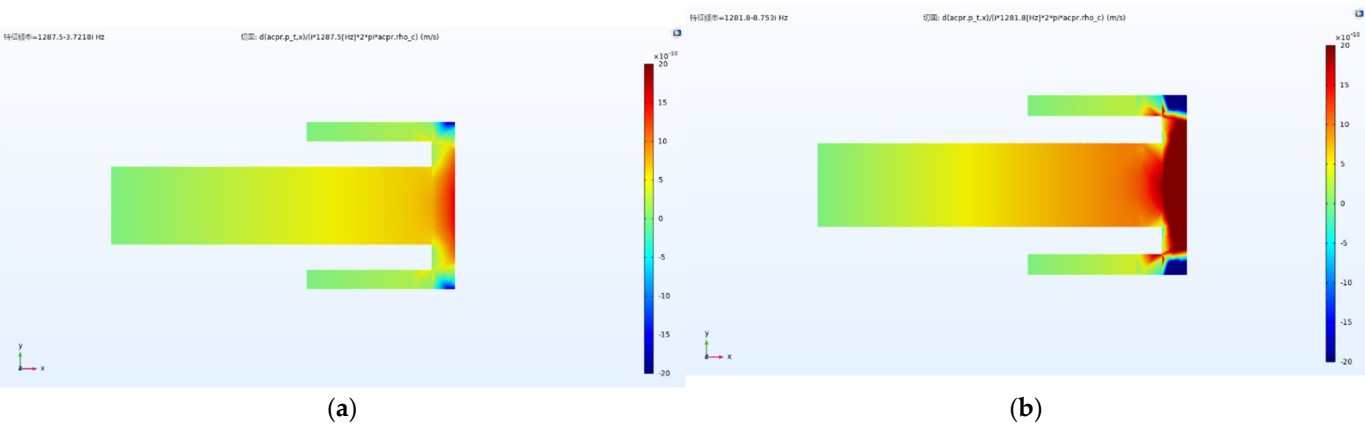

**(a)**          **(b)**

**Figure 6.** Velocity pulsation distribution in injector. (**a**) Scheme 1. (**b**) Scheme 2.

### 4.1.2. Influence of Disturbance Amplitude on Combustion Stability

Schemes 1, 3, and 4 applied flow rate oscillations with amplitudes of 10%, 20%, and 30% to the oxidant circuit, and the flow rate distributions were the same, respectively. Figure 7 shows the heat release rate distribution of Scheme 1. It can be observed that the heat release rate is concentrated in the head region of the combustion chamber, with a certain continuation along the longitudinal direction.

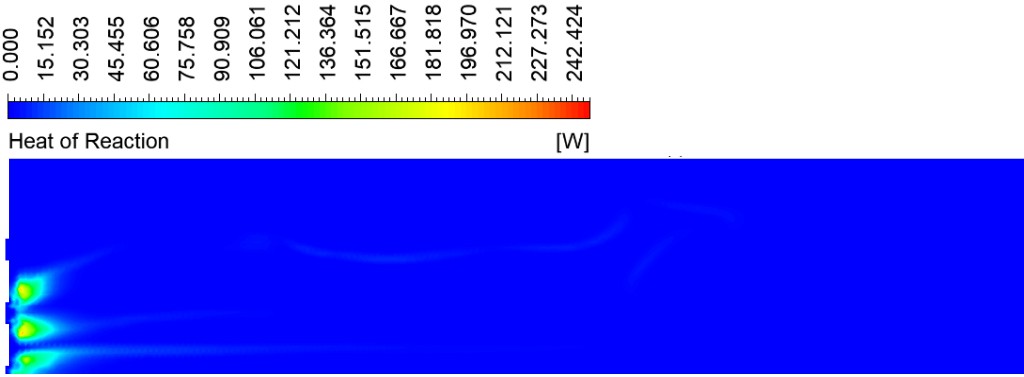

**Figure 7.** The distribution of combustion heat release rate in x-y plane.

The heat release rate distribution of each injector within 2 ms is extracted, and then the effective parameters are obtained according to the flame transfer function introduced in Section 2.1.2. This duration is chosen to cover at least one full cycle of the three frequency simulations. Figure 8a,b are, respectively, the amplitude–frequency characteristic curve and the phase–frequency characteristic curve of flame transfer function, taking the central injector as an example.

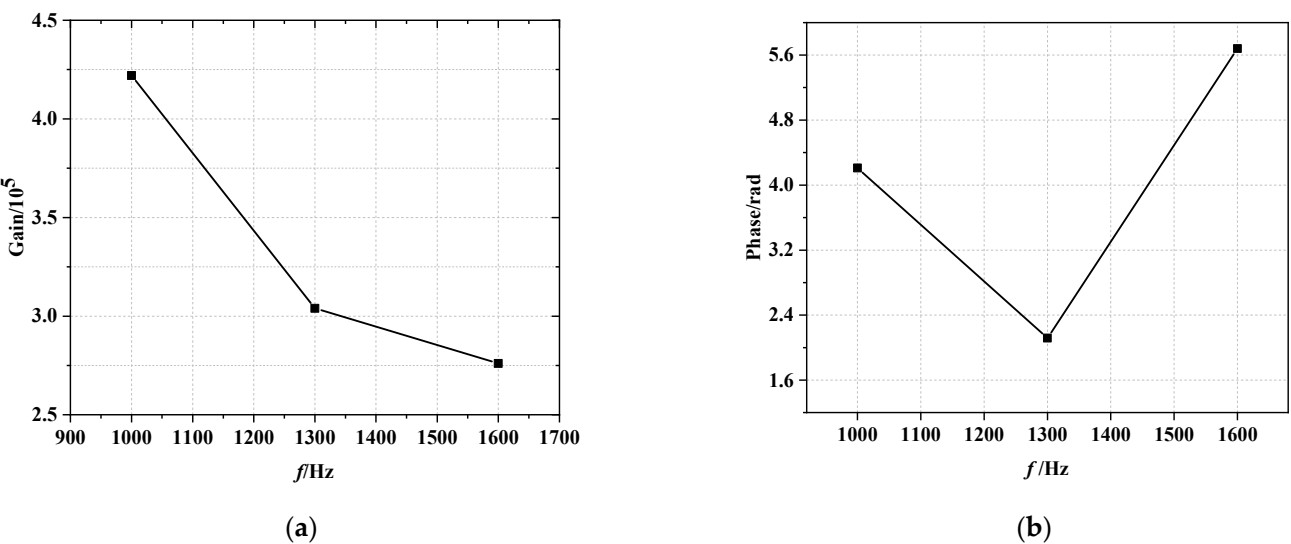

(a)

(b)

**Figure 8.** Central injector flame transfer function. (**a**) Amplitude–frequency curve. (**b**) Phase–frequency curve.

In order to solve the sound field, the simplified geometry of combustion chamber, injectors and the geometric model with compact flame are assembled as shown in Figure 9a. The geometric model is divided by unstructured mesh. After meshing, the mesh of the model consists of 538,367 tetrahedrons. The acoustic solution of the entire combustion chamber is shown in Figure 9b.

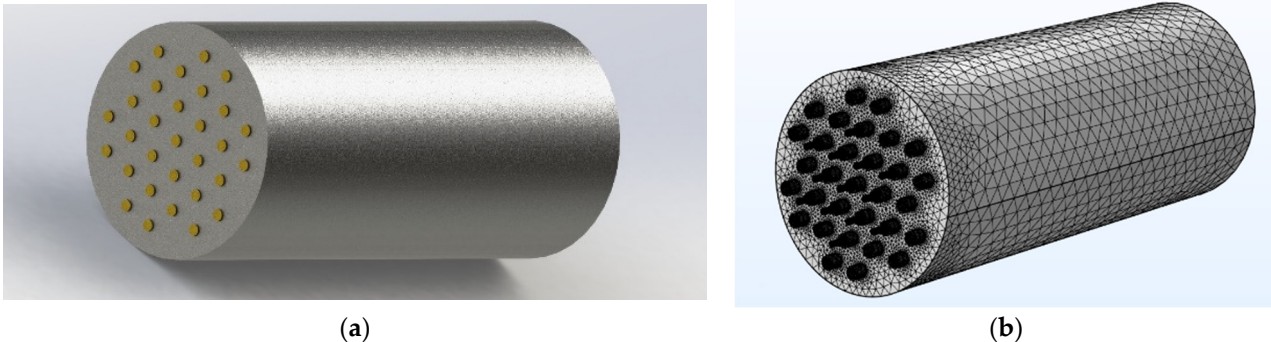

**Figure 9.** (**a**) Geometry of a compact flame. (**b**) Unstructured mesh of the acoustic field solution.

Since the natural frequency of the combustion chamber mainly depends on the combustion chamber structure and the thermodynamic parameters of medium inside, the flame disturbance (or the pulsation of heat release rate) has little effect on the frequency of the acoustic mode, and mainly affects the stability of the acoustic mode. The preliminary calculation of the simplified mode in Section 4.1.1 shows that the first-order longitudinal mode of the combustion chamber is basically around 1280.1 Hz. In this work, the frequency mentioned above is used as the initial frequency of the iteration for the acoustic mode frequency domain simulation so as to shorten the calculation time and improve efficiency.

Schemes 1, 3 and 4 are respectively processed as above, and then the solutions of the Helmholtz equation are obtained, as shown in Table 6. It can be found that with the increase in the amplitude, the eigenfrequency of the first-order longitudinal mode remains basically unchanged, while the growth rates, indicating the stability and the strength of the stability, are all less than zero. With the increase in the amplitude, the absolute value of the growth rate keeps increasing. These phenomena indicate that the increase in the oxidant inlet flow rate oscillation amplitude has an enhanced effect on the combustion instability in this combustion chamber.

**Table 6.** Frequency and stability of first-order longitudinal, zero-order tangential, zero-order radial modes with different disturbance amplitudes.

| $\alpha$ | Eigenvalue | Modal Frequency | Modal Growth Rate | Stability Characteristic |
|---|---|---|---|---|
| 0.1 | $1281.8 - 6.6328i$ | 1281.8 | $-6.6328$ | unstable |
| 0.2 | $1274.2 - 14.547i$ | 1274.2 | $-14.547$ | unstable |
| 0.3 | $1266.9 - 20.363i$ | 1266.9 | $-20.363$ | unstable |

In order to verify the flow rate amplitude influence on the combustion stability of the stable combustion chamber, a variety of amplitude predictions are also carried out for the known stable schemes. The results show that the absolute value of the growth rate also increase gradually with the increase in the amplitude, and the stability characteristics remain unchanged. These phenomena show that the increase in the disturbance amplitude will increase the absolute value of the modal growth rate, that is, the stable combustion condition is more stable, while the combustion unstable condition is more unstable. Therefore, in the following combustion instability prediction, 10% amplitude is used for combustion instability prediction.

### 4.2. Influence of Circumferential Flow Rate Distribution on Combustion Instability

In order to explore the influence of the oxidant circumferential flow rate gradient on the combustion instability, three schemes of different circumferential flow rate distribution are proposed in Section 3.3.2 (Schemes 1, 5 and 6). Scheme 5 increases the propellant one of the two adjacent injectors flow rate by 5% and reduces the other by 5% (Scheme 6 is 10%)

to obtain the circumferential unevenness of flow rate so as to explore the effect of oxidant circumferential flow rate gradient on combustion instability.

Due to the circumferential unevenness of the flow rate distribution, it maintains the center symmetry, but the minimum simulation unit is 1/3 of the combustion chamber. This section uses 1/3 combustion chamber geometry model for simulation with a grid of 4 million. The combustion field steady simulation of Schemes 5 and 6 are carried out to obtain various thermodynamic parameters, and the steady state frequency is roughly estimated according to Equation (20). Since the flow rate distribution does not change much and the structure of the combustion chamber is the same, the steady state characteristic frequency is similar to that of Scheme 1. Therefore, according to the steady state characteristic frequency, the inlet flow rate oscillations of 1000 Hz, 1300 Hz and 1600 Hz frequency are also selected. Figure 10 takes Scheme 5 as an example to show the relationship between the sum of heat release rate in the computational domain and the inlet flow rate pulsation at three frequencies. The processing method described in Section 4.1.2 is used for calculation, and the characteristic frequency and growth rate of the three schemes are shown in Table 7. From Scheme 1 to Scheme 5 and Scheme 6, the circumferential flow rate distribution changes from uniform to uneven, and the combustion of the three schemes are all unstable, with the absolute value of the growth rate being greater than that of Scheme 1, but the combustion instability does not increase with the aggravation of unevenness. Therefore, the circumferential unevenness of the flow rate distribution is unfavorable to the combustion stability in this combustion chamber, while the influence of the circumferential unevenness of the flow rate distribution on the stability is not monotonic.

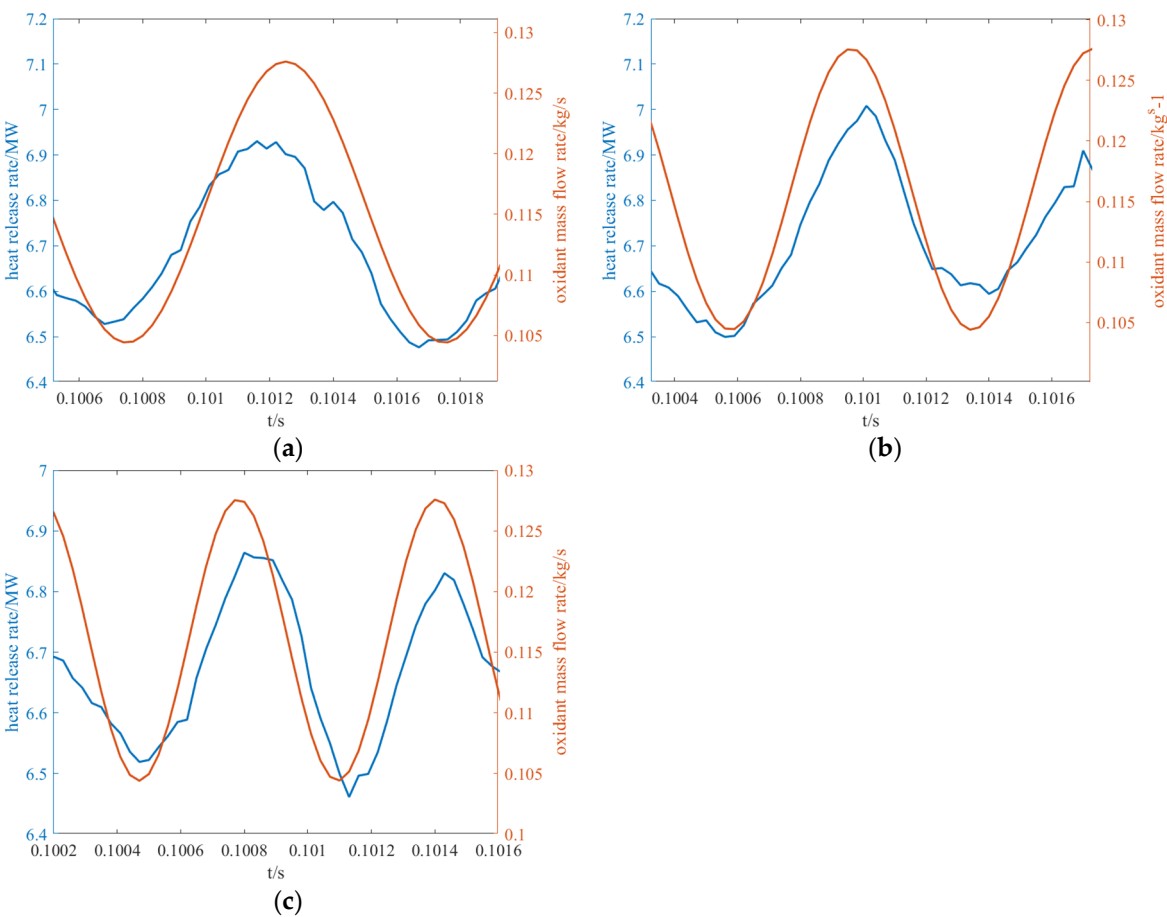

**Figure 10.** The relationship between the sum of heat release rate and inlet flow rate pulsation. (**a**) 1000 Hz. (**b**) 1300 Hz. (**c**) 1600 Hz.

**Table 7.** Frequency and stability of the first-order longitudinal, zero-order tangential, zero-order radial modes for Schemes 1, 5, and 6.

|  | Eigenvalue | Modal Frequency | Modal Growth Rate | Stability Characteristics |
|---|---|---|---|---|
| Scheme 1 | $1281.8 - 6.6328i$ | 1281.8 | $-6.6328$ | Unstable |
| Scheme 5 | $1302.6 - 60.52i$ | 1302.6 | $-60.52$ | Unstable |
| Scheme 6 | $1266.9 - 20.363i$ | 1266.9 | $-20.363$ | Unstable |

*4.3. Influence of Oxidant Radial Flow Rate Gradient on Combustion Instability*

In order to explore the influence of the oxidant radial flow rate gradient on the combustion instability, three schemes of different radial flow rate gradients are proposed in Section 3.3.2 (Schemes 1, 7 and 8), which determine the different ratios of the oxidant flow rate between the inner ring and the outer ring. The radial flow rate gradient of Scheme 7 is the smallest and that of Scheme 8 is the largest.

Steady-state simulations for Schemes 7 and 8 are carried out, respectively. Various thermodynamic parameters and the rough estimated frequency based on steady-state results are obtained according to Equation (20). Similarly, the steady-state eigenfrequencies are similar to those of Scheme 1, so that the inlet flow rate oscillation frequency is set as 1000 Hz, 1300 Hz, and 1600 Hz. Figure 11 takes the central injector as an example to show the amplitude–frequency and phase–frequency curves of the flame transfer function of Schemes 7 and 8, respectively. According to the method described in Section 4.1.2, the characteristic frequency and growth rate of the three schemes are obtained as shown in Table 8. According to the data in the table, it can be found that with the increase in flow rate gradient, Scheme 8 maintains unstable combustion as the basic flow Scheme 1, and its absolute value of growth rate increases, indicating that, compared with Scheme 1, Scheme 8 is more unstable. However, Scheme 7 undergoes stable combustion by reducing the flow rate gradient. Therefore, for this combustion chamber structure, within a certain range, the smaller the flow rate gradient, the more uniform the flow rate of the oxidant circuit, and the more favorable it is to the stability of the stable combustion.

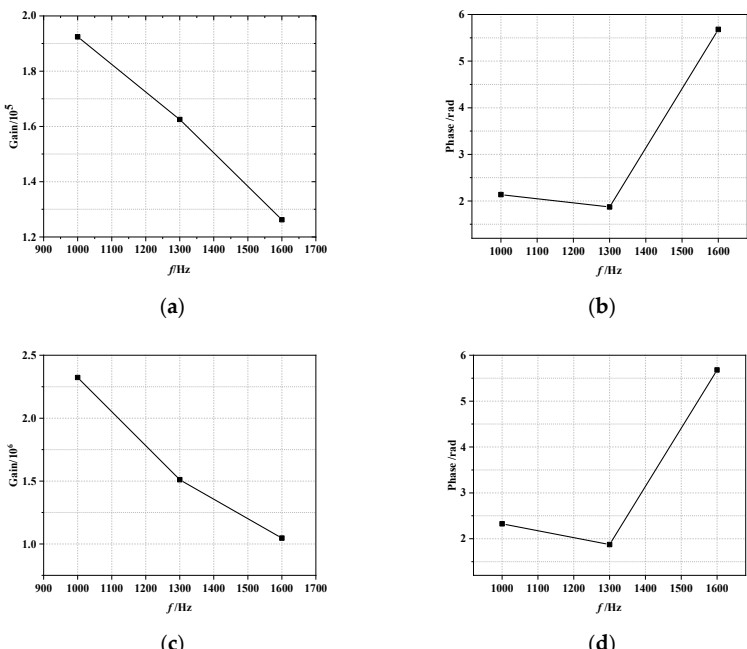

**Figure 11.** The flame transfer function of the central injector. (**a**) The amplitude–frequency curve of Scheme 7. (**b**) The phase-frequency curve of the Scheme 7. (**c**) The amplitude–frequency curve of Scheme 8. (**d**) The phase–frequency curve of the Scheme 8.

**Table 8.** Frequency and stability of the first-order longitudinal, zero-order tangential, zero-order radial modes for Schemes 1, 7, and 8.

|  | Eigenvalue | Modal Frequency | Modal Growth Rate | Stability Characteristics |
|---|---|---|---|---|
| Scheme 1 | 1281.8 − 6.6328i | 1281.8 | −6.6328 | Unstable |
| Scheme 7 | 1332.2 + 218.96i | 1332.2 | +218.96 | Stable |
| Scheme 8 | 1337.5 − 16.439i | 1337.5 | −16.439 | Unstable |

## 5. Conclusions

In this paper, the combustion instability of 10 different schemes are predicted, and the following conclusions are drawn:

The increase in the disturbance amplitude will increase the absolute value of the modal growth rate. The stable combustion will be more stable when increasing the flow rate amplitude, while the unstable combustion will be more unstable when increasing the flow rate amplitude.

The circumferential unevenness of the flow rate distribution is unfavorable to the stability of the combustion, but the influence of the circumferential unevenness of the flow rate distribution on stability is not monotonic.

For this combustion chamber structure, within a certain range, the smaller the oxidant radial flow rate gradient is, the more favorable it is to the stability of the combustion.

**Author Contributions:** Investigation, Y.G., B.Z. and J.C.; methodology, J.L.; software, B.Z. and J.C.; validation, Y.G.; formal analysis, B.Z. and J.C.; resources, Q.F.; data curation, B.Z. and J.C.; writing—original draft preparation, B.Z. and J.C.; writing—review and editing, B.Z., J.C. and Q.F.; visualization, J.C.; supervision, Y.G. and Q.F.; project administration, Y.G. and Q.F. All authors have read and agreed to the published version of the manuscript.

**Funding:** This work was supported by the National Natural Science Foundation of China [Grant Nos. 11922201, 12272026 and U1837211].

**Institutional Review Board Statement:** The study did not require ethical approval.

**Informed Consent Statement:** Not applicable.

**Data Availability Statement:** The data that support the findings of this study are available from the corresponding author upon reasonable request.

**Conflicts of Interest:** The authors declare that there is no conflict of interest regarding the publication of this paper.

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
