# Peer review of "Influence of Flow Rate Distribution on Combustion Instability of Hypergolic Propellant"

_aerospace, doi:10.3390/aerospace9100543_

Round 1

Reviewer 1 Report

The paper investigates the influence of the flow rate distribution on combustion instabilities in a hypergolic model combustor. Generally, the results are relevant and original. Combustion instabilities are a major concern of combustor design and their control through flow rate modulation appears to be a reasonable idea.

It would strengthen the conclusions of the paper, if the impact of the different flow rate distributions on the combustion efficiency would have been quantified. This can be obtained from the preformed CFD analysis and should be added in the paper.

The general analysis relies on a rather simple engineering approach by combining URANS and a basic combustion model with Helmholtz acoustic analysis. It would have been desirable to use a more elaborated model such as LES. Nevertheless, the applied methodology appears to be adequate to identify at least qualitative trends.

However, the paper needs some revisions before publication:

(1)    Add info about the effect of investigated flow rate distributions on the combustor performance (see above).

(2)    The Introduction generally represents a good review of the problem. In line 86 it is mentioned that [18] used LES to look at instability of the “mixed fuel” injection. This work investigates diffusion flames. “mixed fuel” is misleading in this context.

(3)    Line 116: It is mentioned that “the calculation model in this work is too difficult”. It is not clear what is meant by this.

(4)    Line 126 and equation (6): Both, the text and the equation feature a “custom source term”, S. It is not clear what this has been used for. The description of the method and the equations should be related to what was actually be applied.

(5)    In line 170, the assumption of mixing controlled combustion is mentioned (basis of the eddy dissipation model). Can this be justified for the present fuel/oxidizer combination.

(6)    Equation (11): This equation uses the turbulent dissipation rate, epsilon. This is not part in the SST model. Would be clearer the substitute epsilon with omega and k to describe what has actually been done.

(7)    Line 192: Abbreviations are used without prior explanations. R-K should probably be Redlich-Kwong, SRK Suave-Redlich-Kwong and so forth. References are missing for the relations in equations 13 and 14.

(8)    Discussion of the thermodynamics around line 190: Here a statement is missing if the present combustor operates in the super- / trans- or subcrictical range.

(9)    Discussion after line 235: It appears that the “nozzles” in the text are the injectors. (Renaming would make it easier for the reader).

(10)A Reference for the decoupling criterion in line 242 is missing.

(11)Section 1.1.2, line 278: A justification for the perfect gas assumption is missing. Used because in the flame region the gas behaves like a perfect gas ?

(12)Section 2.3, line 355: Would be nice to add a reference to Table 2. “and so on” should be replaced with a more precise list of what has been investigated (or deleted, is there anything else than injection schemes and amplitudes ?)

(13)Table 3 and elsewhere: What is the difference between oxidizer and oxidant ?

(14)Line 418 and conclusions: Reference is made to “hot tests”. But no information is given. Since the comparison to the test is also part of the conclusions, it is important to provide a reference to the experimental data and provide the results used for the discussion in the present paper.

(15)Figure 7: It appears that (a) to (e) are exactly the same pictures, the legend should be in W/m3 instead of W ? Generally (Figs 7,8 10, the text in the figures can hardly be read).

(16) Conclusions: Again, it is mentioned that the results are consistent with the test without any information about the test. Hence, the statement is not useful.

(17)The sentence “it is more stable under stable conditions and more unstable under unstable conditions” is hard to understand. A more precise statement would be helpful for the reader.

The English needs minor improvement (e.g. shew = showed)

Reviewer 2 Report

This manuscript deals with the characterization of combustion instability in a Liquid Propellant Rocket Engine (LRE) operating on hypergolic propellant. The approach used by the authors is based on extracting the flame transfer function from the solution of the decoupled combustion-acoustic system. The manuscript fits the scope of Aerospace but needs major revision. My detailed comments are given below.

(1) The authors' English must be thoroughly improved in terms of both spelling and grammar.

(2) A list of abbreviations and nomenclature must be provided. Some abbreviations are not identified (like “SRK and PR equations, and is simpler than the EXP-RK and PR-EXP equations.”)

(3) The literature review is incomplete. The literature on combustion instability in LRE is very rich with different approaches, which are worth to be mentioned. For example, in the recent 15 years, there were many publications in the EUCASS Book Series "Progress in Propulsion Physics."

(4) The authors' approach is described very chaotically and confusingly. It must be rewritten in a systematic way.

(5) The sentence in Lines 181-182 (“As long as k/e>0 is established, the reaction will proceed, with no ignition source required to start the combustion reaction”) needs explanation. The ED-model was originally suggested for premixed combustion and its use for self-ignition phenomena is not substantiated.

(6) The real-gas equation of state (12-14) must be validated. The authors must demonstrate its applicability to the research conditions.

(7) On the one hand, the authors speculate about the importance of the real gas equation of state, but on the other hand, they use perfect gas relationships when deriving the transfer function, which is misleading.

(8) There is no analysis of errors accompanying the authors' approach. The error could be estimated by providing a simple example illustrating the approach. Such an example must show the influence of the combustion-acoustic system decoupling by comparing with the results obtained by coupled calculations. 

Round 2

Reviewer 1 Report

Thank you for adequatley adressing the review comments. It would still be nice to reformulate lin 440 with the reference to the classified hot tests and mention something like "the dominance of the first mode was also observed in ground tests carried out by....".

Reviewer 2 Report

The authors have partly addressed my comments by adding some additional references and explanations but not proving the applicability of their approach at a simple example. Unfortunately, the structure of the manuscript was not changed and it is still hardly readable. Nevertheless, the results reported by the authors could be of interest for journal readers. Therefore, the manuscript could be considered now for publication.